# Presence of sodefrin precursor-like factor pheromone candidates in mental and dorsal tail base glands in the plethodontid salamander, *Karsenia koreana*

**Jared H. DeBruin[1]\*, Damien B. Wilburn[2,3], Richard C. Feldhoff[4], Nancy L. Staub[1]**

**1** Department of Biology, Gonzaga University, Spokane, Washington, United States of America, **2** Department of Genome Sciences, University of Washington, Seattle, Washington, United States of America, **3** Department of Chemistry and Biochemistry, The Ohio State University, Columbus, Ohio, United States of America, **4** Department of Biochemistry and Molecular Biology, School of Medicine, University of Louisville, Louisville, Kentucky, United States of America

\* jaredhdebruin@gmail.com

**Data Availability Statement:** All relevant data are within the paper.

## Abstract

Plethodontid salamanders are well known for their distinct courtship rituals and the associated pheromonal signaling. However, little is known about pheromones produced in the lone Asian plethodontid species *Karsenia koreana*. Here, we examined the localization patterns of proteins of the sodefrin precursor-like factor (SPF) pheromone system in *K. koreana*. Using an antibody generated against SPF proteins from another plethodontid, *Desmognathus ocoee*, we tested three types of skin glands in *K. koreana* males via immunohistochemistry: the mental gland and two types of dorsal tail base glands–caudal courtship glands and dorsal granular glands. SPF immunoreactivity was detected in the known courtship gland, the mental gland, as well as granular glands, but not in caudal courtship glands. Due to immunoreaction specificity, we hypothesize the proteins of the SPF system in *K. koreana* and *D. ocoee* are structurally and functionally related and are used as courtship pheromones in *K. koreana*. Also, we hypothesize that *K. koreana* males transmit SPF to the female during the tail-straddling walk via dorsal granular glands. Finally, *K. koreana* male caudal courtship glands may be producing SPF proteins that are not recognized by our SPF antibody or these glands may play a different role in courtship than anticipated.

## Introduction

Pheromones, chemical signals that produce a response in members of the same species [1], are used throughout the tree of life. Identifying which signals elicit which behavioral or physiological responses is of great interest [2–4]. To confirm that a substance is a pheromone, however, can be a challenge. Since pheromones are often released as a part of chemical mixtures, isolating the signal, its receptor, and measuring the effects of a signal, can be difficult [5–8]. As a result, only a few pheromone-receptor pairs have been described [9]. Some examples of

**Funding:** The funding received for this study is outlined as follows: NLS, no grant number (internal funding), Gonzaga Science Research Program Fund, https://www.gonzaga.edu/academics/centers-institutes/center-for-undergraduate-research-and-creative-inquiry/gonzaga-science-research-program DBW, NICHD R00-HD090201, National Institute of Child Health and Human Development, https://www.nichd.nih.gov/grants-contracts The funders had no role in study design, data collection and analysis, decision to publish, or preparation of the manuscript.

**Competing interests:** The authors have declared that no competing interests exist.

pheromone-receptor pairs can be found in insects, where pheromones are used for mate selection, food localization, predator warning signals, and more [10–12].

Because of the difficulty in identifying pheromones, chemical signals that scientists hypothesize to be pheromones are aptly named "pheromone-candidates." One example in frogs is the sodefrin precursor-like factor (SPF) family of proteins [13]. These signals were described as pheromone candidates in frogs because they are known to be pheromones in salamandrids and plethodontids [14–16], but their function in frogs is, as of yet, unknown [13].

Behavioral evidence indicates that SPF proteins are involved in courtship in salamandrids and plethodontids [14]. Courtship is defined as behaviors that maintain reproductive actions between mating partners; it does not refer to initial mate attraction [17]. The courtship pheromone SPF was identified in *Desmognathus ocoee*, a species of plethodontid salamander [14]. During courtship, a male *D. ocoee* will scratch the female's dorsal skin with hypertrophied teeth and rub over these scratches with his submandibular region [14, 18, 19]. In *D. ocoee* and in other plethodontids, the submandibular region of the male contains a group of exocrine glands called the mental gland [14, 20]. Within the individual secretory glands, cells secrete substances into the gland lumen for the eventual release from the gland's secretory duct [15, 21]. SPF proteins are present in the mental gland tissue of *D. ocoee* [14]. When a proteinaceous extract, that is primarily SPF proteins, is applied to the dorsal skin of female *D. ocoee*, female receptivity of courtship behavior increases [14]. Receptivity refers to the female's "acceptance" of the courtship behavior, quantified as a decrease in courtship duration [14]. The target organ of this transdermal-based method of pheromone delivery is unknown. However, the proteinaceous extract ellicited a distinct behavioral response in females, which resulted in classifying the major components of the fraction, SPF proteins, as pheromones [14, 22].

Interestingly, SPF proteins are linked to multiple courtship behaviors in plethodontids. For example, in addition to being associated with courtship behaviors involving the mental gland in *D. ocoee*, recent studies have detected SPF mRNA in dorsal tailbase glands in other plethodontids [23]. These glands, named caudal courtship glands after their hypothesized function [23], are found on the dorsal tail base of male plethodontids and are morphologically similar to the mental gland [24, 25]. Additionally, caudal courtship glands and the mental gland react similarly to periodic-acid Schiff, a histochemical reaction that detects neutral carbohydrates [26]. While evidence documents SPF proteins' involvement in salamander (and other amphibian) courtship, more behavioral studies are required to identify SPF as a courtship pheromone in other behaviors and species.

Our goal was to examine localization patterns of SPF proteins in the mental gland and in caudal courtship glands of the plethodontid Korean crevice salamander, *Karesenia koreana*, using an antibody against *D. ocoee* SPF. Additionally, we examined granular tail base glands which are found adjacent to caudal courtship glands. We hypothesized that SPF immunoreactivity would be observed in the mental gland and caudal courtship glands, but not in the dorsal granular glands. Currently, there are over 500 species of plethodontid salamanders [27] and *K. koreana* is a relatively recent discovery from 2005 [28]. This species is the only plethodontid salamander native to Asia [28] and has sparked biogeographical, phylogenetic, ecological, cytogenetic studies [29–32]. While the morphology of skin glands of *K. koreana* has been examined [30], the putative pheromones produced by skin glands have not.

## Materials and methods

### Specimen retrieval, dissection, sectioning, and mounting

Tissue from three *K. koreana* and three *D. ocoee* male specimens were received from the private collection of D. R. Vieites (DRV 5558, DRV 5551, DRV 5555) and S. J. Arnold (SJA41356,

SJA41357, SJA41358) that were fixed in 10% formalin and stored in 70% ethanol. The dorsal tail base and submandibular region were dissected, embedded in paraffin (Paraplast Plus, Fisher Scientific), and sectioned at 8–10 micrometers by a rotary microtome (Lecia 2035 Jung Biocut Microtome). Sections were mounted onto Fisher Superfrost Plus microscope slides for staining and immunohistochemistry. Standard histological procedures were used [33].

## Quad staining methods

Caudal courtship glands, granular glands, and mental glands were identified histologically in *K. koreana* using the Quad stain adapted from Floyd [34] and Staub and Paladin [35]. The Quad stain consists of periodic-acid Schiff (PAS) to identify neutral carbohydrates, napthol yellow to identify proteins, Alcian blue (pH = 2.0) for mucopolysaccharides, and methyl green for nuclear DNA. For the PAS reaction, Schiff specificity was tested by treating tissues without periodic acid or with periodic acid followed by dimedone for 1 hour at 60°C. Dimedone blocks the aldehydes produced by the reaction of carbohydrates with periodic acid, preventing the Schiff reagent from reacting with them [36]. Mental and caudal courtship glands are strongly positively for PAS; granular glands are expected to be negative or just slightly positive for PAS and stain positively for napthol yellow [37].

## SPF antibody purification

Antisera to *D. ocoee* mental gland proteins was prepared by immunizing two rabbits with proteins extracted from *D. ocoee* mental glands following the methods from Houck et al. [14]. To enrich for SPF-specific antibodies, recombinant antigen was prepared using recombinant expression methods adapted from Wilburn et al. [38] and antibody purification methods from Wilburn and Feldhoff [39]. Briefly, *D. ocoee* SPF I-01 cDNA with a N-terminal 6xHis tag was cloned into the pET45b expression vector (EMD-Millipore), transformed into Rosetta2 *E. coli* cells (EMD-Millipore), transformed plasmids validated by Sanger sequencing, and recombinant SPF expressed by the addition of 100 μM IPTG to mid-log phase cultures for 3 hours. Because *E. coli* are not able to naturally fold proteins with large amounts of disulfide bridges such as SPF, recombinantly expressed SPF accumulated in inclusion bodies that were harvested by centrifugation following cell lysis [40]. Inclusion bodies were solubilized with 8M urea deionized with Rexyn I-300 beads (Sigma-Aldrich), disulfide bonds reduced by addition of 50 mM DTT for 30 minutes, and alkylated by incubation with 100 mM iodoacetamide in the dark for 45 min. Insoluble material was removed by centrifugation, and denatured recombinant SPF purified using Ni-NTA resin (Pierce) with all buffers containing deionized 8M urea to maintain SPF solubility. Recombinant SPF was confirmed to be >95% pure by SDS-PAGE. A recombinant SPF antigen column was prepared by incubation of ~1mL CDI activation of CL-6B agarose beads (Sigma) with recombinant SPF that was buffer exchanged into freshly deionized 8M urea (to ensure removal of potential free $NH_3$ that would compete for bead coupling) that was then supplemented with 100mM $NaCO_3$, pH 10. The slurry was mixed overnight at 4°C before being packed into a column and blocked with > 10 mL 100mM Tris, pH 8. SPF antibodies were purified by several iterations of incubating 1mL *D. ocoee* mental gland antisera with the resin at 4°C overnight, washing the column with 10 mL 500 mM NaCl/0.05% Tween-20/20mM Tris, pH 8, and eluting antibodies with 3 mL 100 mM Glycine, pH 3 that was quickly neutralized by addition of 1 mL 1 M $Na_2PO_4$. Multiple preparations of anti-SPF were pooled, concentrated, and buffer exchanged to 1X Phosphate Buffered Saline (PBS) using a 30 kDa centrifugal ultrafilter (Millipore).

## Immunohistochemical staining methods

Immunohistochemistry using the antibody against *D. ocoee* SPF was used to test for the presence of SPF in *K. koreana* tissue [40]. Mounted tissue sections were heated for 60 minutes at 60°C to ensure tissue sections adhered to slides. Standard histological methods were used for deparaffinization and hydrating sections [33]. For antigen retrieval, tissue sections were placed in citrate buffer (pH 6) at 70°C for 30 minutes and washed in PBST 5 times (PBS with 0.05% Tween-20). To block excess proteins, sections were incubated with normal goat serum (Fisher Scientific Ultra-Sensitive ABC Rabbit IgG staining kit or Vector Labs Elite ABC kit).

The primary antibody was applied to tail base or mental gland tissue sections in 1:1,000 dilutions (in PBST) and incubated for 1–2 days. After incubation, slides were washed 5 times with PBST and incubated with the biotinylated secondary antibody for 30 minutes. For detection, streptavidin bound HRP chemistry was utilized with NOVARed and metal enhanced DAB substrates (Vector Biolabs and ThermoFisher). Gill's hematoxylin or methyl green was used as a counterstain. Negative primary antibody controls were used to assess levels of non-specific and background staining.

## Image collection and processing

Observations were made using a Leica DME light microscope. Images of sections were taken using an EOS Rebel 5 camera. White balance was adjusted in Fiji.

## Results

We identified the three gland types in *K. koreana*–the mental gland, caudal courtship glands, and dorsal granular glands using the Quad stain and existing literature [30, 34, 35, 37]. We examined the localization patterns of SPF proteins in these glands using immunohistochemistry. *Karsenia koreana* and *D. ocoee* mental glands exhibit immunoreactivity with the SPF antibody (Fig 1) compared to controls (Fig 2), shown by deep red staining in the cytosol of the secretory cells. Male *K. koreana* possess caudal courtship glands on the dorsal tail base (Fig 3). The cytosols of their secretory cells did not exhibit SPF immunoreactivity, as shown by lack of red staining (Fig 3). Finally, granular glands were identified with naphthol yellow positive and

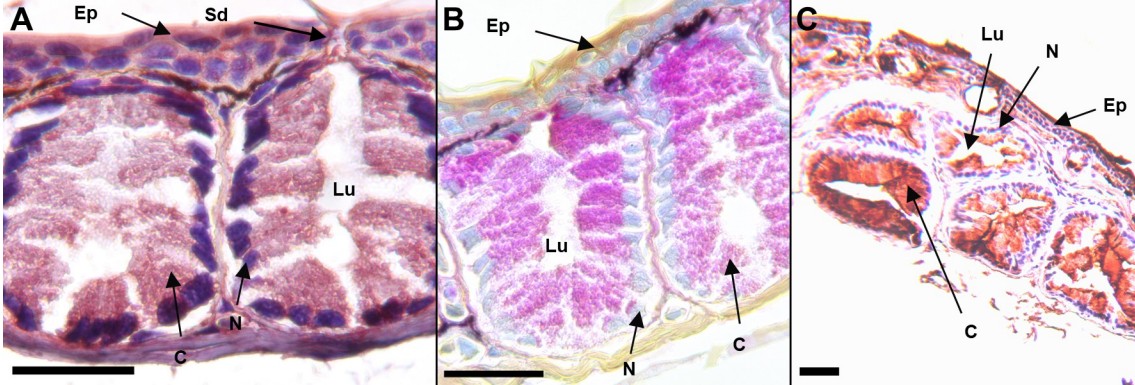

**Fig 1.** Mental gland of *K. koreana* (A, B) and *D. ocoee* (C). The mental gland is an aggregate of simple exocrine glands. Secretory cells line the periphery of the gland and contain a granular product. The cytosol of *K. koreana* secretory cells are positive for SPF immunoreactivity, indicated by a dark red colored product (A). The antibodies were made against SPF proteins from *D.ocoee* mental glands. *Desmognathus ocoee* mental gland tissue had SPF immunoreactivity as expected (C). The mental gland is PAS positive as well, indicated by the magenta reation product in the cytosol of the secretory cells and for secretory products in the gland lumen (B). Results were consistent between inidividuals (n = 3 for each species). Scale bars are 100 μm. N = nucleus; C = cytosol; Lu = lumen; Sd = secretory duct; Ep = epidermis.

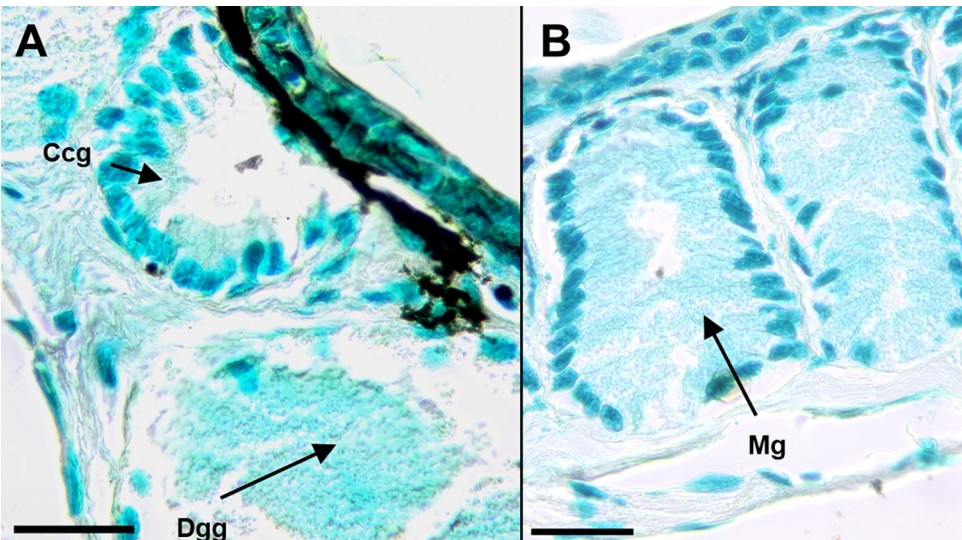

**Fig 2.** Negative controls for *K. koreana* caudal courtship glands (A), and mental gland (B), without SPF primary antibody. Negative controls, treatments without the primary antibody, were used to assess background and non-specific staining. No dark red colored product is visible in these controls indicating the absence of non-specific staining from the secondary antibody. Methyl green was used as a nuclear stain to identify cells. This negative control treatment was used to assess background and non-specific staining. The arrows point to the secretory cells within the caudal courtship (A) and within an individual gland of the mental gland. Scale bars are 100 μm, n = 3. Ccg = caudal courtship gland, Dgg = dorsal granular gland, Mg = one individual gland within the mental gland.

PAS negative secretory cells (Fig 2), consistent with previous literature. The cytosols of these cells are also granular in appearance. The secretory cells of granular glands show immunoreactivity with the SPF antibody compared to controls (Figs 2, 3).

## Discussion

We identified two gland types in *K. koreana*, the mental gland and dorsal granular glands, that were immunoreactive to the SPF antibody (made against *D. ocoee* SPF proteins) (Figs 1 and 2).

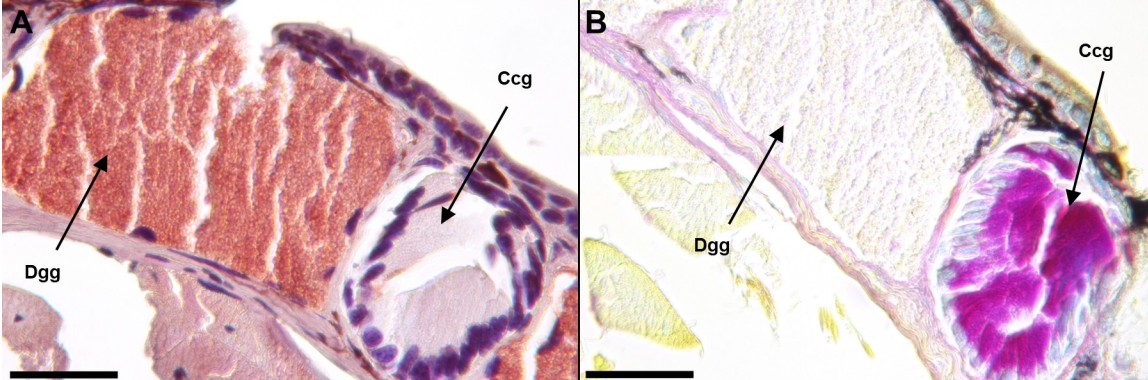

**Fig 3. Caudal courtship and dorsal granular glands in *K. koreana* tail base tissue.** Secretory products in dorsal granular glands are positive for SPF immunoreactivity, indicated by a dark red colored product (A). In contrast, the secretory cells of caudal courtship glands are negative for SPF immunoractivity, when tested with antibodies made against SPF proteins from the *D.ocoee* mental gland (A). Interestingly, the caudal courtship glands are positive for PAS (magenta in color (B)), while the dorsal granular glands are negative (B). Caudal courtship glands are similar in color to mental gland treated with PAS (Fig 1B). Results were consistent among individuals (n = 3). Scale bars are 100 μm. Ccg = caudal courtship gland, Dgg = dorsal granular gland.

While the skin glands of *K. koreana* have been previously described [30], this is the first report of localization patterns of SPF proteins in glands of this species. SPF mRNA has been reported in caudal courtship glands of plethodontids previously [23], but contrary to our prediction, *K. koreana* caudal courtship glands were not immunoreactive to the SPF antibody (Fig 3) when compared to negative controls (Fig 2).

SPF proteins are present in *K. koreana* mental glands, suggesting that male *K. koreana* use SPF proteins to increase female receptivity during courtship. While the courtship behavior of *K. koreana* has not yet been observed, the transdermal method of protein delivery is ancestral to its lineage [27, 41, 42]. Thus, we predict that *K. koreana* uses a transdermal delivery system, similar to *D. ocoee*. *Karsenia koreana* may also use other methods of pheromone delivery. *Karsenia koreana* shares a most recent common ancestor with the *Hydromantes* group [31, 43]. *Hydromantes italicus* males rub their submandibular regions extensively on the female's back, indicative of a transdermal delivery system similar to *D. ocoee* [44]. However, male *H. italicus* occasionally clasp the female's neck and press their mental glands on the female's nares, suggesting an olfactory delivery of mental gland secretions [44]. Also, it is plausible that females may be delivering pheromones to males during courtship [45], although unfortunately we did not include females in our samples. Studies on *K. koreana*'s courtship behavior will be invaluable to understand how *K. koreana*'s mental gland secretions, and secretions from other glands, are used for communication.

Because the SPF antibody made against denatured *D. ocoee* SPF proteins binds to proteins in the mental gland and dorsal granular glands of *K. koreana*, there are perhaps highly conserved regions within these proteins. As SPF sequences substantially vary between species [46], understanding more about these conserved regions and their function across species groups will be most interesting. More studies that examine the glandular distribution of pheromone gene expression [23] and the molecular structure and evolution of pheromones [47] will be critical to increase our understanding of the complexities of pheromone structure, function, and evolution.

That SPF proteins were detected in dorsal granular glands raises questions about their functional significance. These glands are found on the dorsal tail base of males, the area in contact with the female's nares during the tail straddling walk, a sterotypical courtship behavior in plethodontids [41]. Pheromones may be delivered to the female's vomeronasal organ during this stage to help maintain contact and ensure spermatophore pick up by the female [48]. While caudal courtship glands have been found to contain pheromone mRNA [23], this is the first report of dorsal grandular glands containing SPF proteins. We hypothesize that these dorsal granular glands play a role in communicating to the female during the tail-straddling walk. More studies that focus on identifying pheromone-candidates in granular glands will be important in determining the functional significance of these glands.

We did not detect SPF proteins in caudal courtship glands, the glands hypothesized to be involved in courtship in other species [26, 49–51]. Phylogenetic analyses indicate that *D. ocoee* and *K. koreana* share a most recent common ancestor from the Late Cretaceous, more than 50 million years ago [52]. Since SPF proteins vary between species and evolve rapidly [46], proteins involved in the SPF protein family in *K. koreana* may be unrecognizable to the *D. ocoee* derived antibody. Alternatively, SPF proteins may not be produced in these caudal courtship glands at all. Other pheromones may be produced or these PAS positive glands may serve a different function in *K. koreana*.

In summary, SPF protein localization patterns suggest that *K. koreana* mental glands secrete SPF proteins during courtship. SPF localization in the dorsal granular glands but not in caudal courtship glands raise questions about the functional significance of caudal courtship and

dorsal granular glands. Observing *K. koreana* courtship behavior and isolating and characterizing glandular secretions will be critical to understanding the function of these glands in courtship.

## Acknowledgments

We thank Angie Hinz for logistical assistance and the reviewers for greatly improving the quality of this manuscript.

## Author Contributions

**Conceptualization:** Jared H. DeBruin, Nancy L. Staub.

**Data curation:** Jared H. DeBruin.

**Formal analysis:** Jared H. DeBruin, Nancy L. Staub.

**Funding acquisition:** Nancy L. Staub.

**Investigation:** Jared H. DeBruin, Damien B. Wilburn, Richard C. Feldhoff, Nancy L. Staub.

**Methodology:** Jared H. DeBruin, Damien B. Wilburn, Richard C. Feldhoff, Nancy L. Staub.

**Project administration:** Jared H. DeBruin, Nancy L. Staub.

**Resources:** Damien B. Wilburn, Nancy L. Staub.

**Supervision:** Nancy L. Staub.

**Validation:** Richard C. Feldhoff.

**Visualization:** Jared H. DeBruin, Nancy L. Staub.

**Writing – original draft:** Jared H. DeBruin, Richard C. Feldhoff.

**Writing – review & editing:** Jared H. DeBruin, Damien B. Wilburn, Nancy L. Staub.

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
