## [Decision Letter · Decision Letter 0]

11 Jan 2023

PONE-D-22-29926Presence of the conserved sodefrin precursor-like factor pheromone in mental and dorsal tail base glands in the plethodontid salamander, *Karsenia koreana*PLOS ONE

Dear Dr. DeBruin,

Thank you for submitting your manuscript to PLOS ONE. After careful consideration, we feel that it has merit but does not fully meet PLOS ONE’s publication criteria as it currently stands. Therefore, we invite you to submit a revised version of the manuscript that addresses the points raised during the review process.

Your manuscript has been assessed by three expert reviewers, whose comments are appended below. As you will see from the comments, the reviewers are broadly positive about the design and execution of the study, but were more critical of the scientific writing. The reviewers have offered detailed suggestions of how the manuscript can be revised to more effectively contextualise and explain your findings - please ensure you respond to each point carefully in your response to reviewers document, and modify your manuscript accordingly.

We look forward to receiving your revised manuscript.

Kind regards,

Dr Joseph Donlan

Senior Editor

PLOS ONE

Journal Requirements:

“We thank Angie Hinz for logistical assistance and the Gonzaga Science Research Program for funding.”

“The funding received for this study is outlined as follows:

NLS, no grant number (internal funding), Gonzaga Science Research Program Fund, https://www.gonzaga.edu/academics/centers-institutes/center-for-undergraduate-research-and-creative-inquiry/gonzaga-science-research-program

DBW, NICHD R00-HD090201, National Institute of Child Health and Human Development, https://www.nichd.nih.gov/grants-contracts

Reviewers' comments:

Reviewer's Responses to Questions

**Comments to the Author**

1. Is the manuscript technically sound, and do the data support the conclusions?

Reviewer #1: Yes

Reviewer #2: Yes

Reviewer #3: Yes

2. Has the statistical analysis been performed appropriately and rigorously? 

Reviewer #1: N/A

Reviewer #2: N/A

Reviewer #3: N/A

3. Have the authors made all data underlying the findings in their manuscript fully available?

Reviewer #1: Yes

Reviewer #2: Yes

Reviewer #3: Yes

4. Is the manuscript presented in an intelligible fashion and written in standard English?

Reviewer #1: Yes

Reviewer #2: Yes

Reviewer #3: Yes

5. Review Comments to the Author

Reviewer #1: This study investigates the production of potential protein pheromones in glandular tissue of the only Asian plethodontid salamander. The authors show via immunohistology that two of three glands produce SPF – an ancient pheromone known from many amphibians.

The manuscript is well written, the experiments are nicely conducted and the results are nicely presented and comprehensively discussed. The methods used by the authors are a very good solution, especially when only preserved tissue is available.

My only larger issue is the structuring of the introduction (which should be relatively easy to fix). Besides that, I only have a few smaller issues/comments.

Main issue:

Introduction:

The introduction starts very abrupt, right away introducing the study species and only then talking about salamander pheromones. I think a bit of restructuring would give the introduction a better flow and also make it more interesting for a broader readership. I suggest starting with a few broader sentences regarding pheromones and chemical communication in general, then go over to amphibian chemical communication, then introduce SPF as the most ancient pheromone found in amphibians, and then go into detail about plethodontid courtship and pheromones, before introducing the species of interest and your hypothesis with regard to this species.

Further comments:

- line 66-68: the way this sentence is written it sounds a bit like SPF is an ancient pheromone that today only exists in plethodontids. I know this is not what you mean, but maybe you can add that it has been shown to have a pheromone function in many salamander families (for review see CSiV-chapter from Bossuyt et al. 2019) and has been even found in many frogs (where, however, proof that it is a pheromone is still lacking)

- line 103: since Richard C. Feldhoff is a co-author in this paper, I think it is not necessary to mention this here (you could instead maybe add a statement at the end of the manuscript stating the different author contributions)

- line 96: …the PAS reaction (stains neutral carbohydrates),…

- line 99: what does dimedone in this context?

- line 104/106-7: …following the methods from Houck (2008) …. adapted from Wilburn et al. (2018) and antibody purification methods from Wilburn and Feldhoff (2019).

- line 158-162: this part is not necessary here

- only when seeing Fig. 2 and reading line 232 in the discussion I realized that the caudal courtship glands and the granular glands are directly adjacent to each other. It would be helpful to already describe their positions better in the introduction or (even better) show a picture/drawing of a salamander where the different glands are marked/pointed out

- 243ff: this might exceed the framing of the discussion, but (maybe for future studies) it could be worth considering that some of the plethodontid glands release volatile pheromones, as shown for example for reed frogs and mantellids?

All in all, this is a very nice study and manuscript and I congratulate the authors on their interesting findings!

Reviewer #2: DeBruin et al., Presence of the conserved sodefrin precursor-like factor pheromone in mental and

dorsal tail base glands in the plethodontid salamander, Karsenia koreana

I enjoyed reading this article, and I think it makes a nice contribution to the understanding of the evolutionary history of plethodontid pheromones. I have mostly minor comments aimed at improving clarity below. On a more major note, you might consider adding some phylogenetic/evolutionary history context to the Discussion. In the Introduction, you argue that the pheromone SPF is conserved within the family, and describe some of the phylogenetic/evolutionary history of this (Lines 64-68), but you do not circle back to the conservation of SPF in the Discussion now that you have discovered it in K. koreana. It would be helpful to put SPFs discovery in K. koreana in context of the entire family. I think it would also help your argument in Lines 243-249 to discuss how the isolation in both evolutionary time and geographic distance between K. koreana and other plethodontids might explain why a D. ocoee SPF anitbody might not pick up a long-diverged SPF variant in the Ccg.

Minor Comments:

Line 49: I might be misremembering, but isn't K. koreana the only extant plethodontid throughout all of Asia? If so, you can make this statement stronger than "the only plethodontid in Korea."

Line 69-70: "another highly conserved behavior" - replace "another" with "a", and add "courtship" before behavior to be clear. You just talked about behaviors with variation, and a conserved pheromone, so tail straddle walk is the first highly conserved behavior you're discussing.

Line 84: add "three male" before Desmognathus, OR add "each" after specimens, to be clear that you had three of each species.

Lines 158-162: I don't know that the "as mentioned previously" is necessary. You could shorten the first two sentences of the Results into one sentence (example below), and combine this paragraph with the next paragraph for a more concise Results section.

Ex. "The skin glands of plethodontid salamanders have been thoroughly described in several species, however characterization in K. koreana has been limited. Here, we examined three skin gland types in K. koreana:..."

Line 181: In the Figure 2 Legend, you are missing close parenthesis after (B

Lines 185-198: Were you expecting the Ccg/Gg to show presence of SPF? That is, do D. ocoee or other plethodontids show SPF in the Ccg/Gg or just in the Mg? Your D. ocoee reference (Fig. 1C.) is just of the Mg. If you did stain Ccg and Gg in D. ocoee, you might add that to Figure 2 to mirror how you laid out Figure 1.

I was initially confused on what Figure 3 was showing until I reread lines 196-198 and the legend a few times. It might be worth adding something to the legend similar to the sentence on Lines 196-198 to make it clear that this is a control for the SPF antibody staining in Figs. 1 and 2. Indeed, you have a very clear statement on the importance of this image at the end of the first Discussion paragraph (Lines 209-210).

Line 241: Change "–could they be related to SPF?" to "and they could be [structurally?] related to SPF". Where I have structurally, put whatever qualifier(s) make sense, since "related" by itself is extremely broad.

Line 247-249: For this next step, you would need live individuals from which to extract the secretions, correct? I would state this outright for the benefit of those who don't work on glandular chemistry (like myself).

Reviewer #3: *General comment:*

Given the straightforward study goal and most of the co-authors that are well known in the field I expected a better written article with little investment in the review process. However, this is not the case. The English is generally at a good level but with strange expressions and statements to the point that I felt like correcting a bachelor or master thesis and not a scientific article submitted to a respectful journal by experts in the field. There is a major oversight of not mentioning the diffusion delivery in plethodontids to provide more context to the study, but this can be easily corrected. Since most of the authors are well-known experts in the research field, I do not see any reason why this study could not have a solid context that it and the journal deserve. As it is written now, I more get the impression that the article is written to ‘get it out there as fast as possible’ than to provide high quality material for the readers interested in the subject.

*In detail:*

TITLE:

The title suggests that SPF is a pheromone in Karsenia salamanders while no such studies have been conducted. Furthermore, although there is indication that SPFs from mental glands act as a pheromone in North American plethodontids (based on 20-25 kDa band of fraction D with unknown content of multiple bands tested on Desmognathus ocoee by Houck et al. in 2008), the SPF mode of action is completely unknown, i.e. where are female receptors, how does SPF reach them and how does it affect female behaviour? The production of SPF in mental glands and potential use of SPF as a pheromone in the tail-base glands in Asian plethodontids is the biggest novelty of this research, but it also calls for caution calling SPFs from tail-base glands pheromones before empirical studies have been conducted. I would suggest calling SPF proteins “pheromone-candidates” instead, something in the sense of:

**“Sodefrin precursor-like factor pheromone candidates present in both mental and dorsal tail base glands in the plethodontid salamander, Karsenia koreana”**

The title indicates ‘tail base glands’ while the abstract mentions ‘dorsal granular glands’ and ‘caudal courtship glands’ which I find quite confusing. There should be consistency in the naming of the tissues throughout the article according to existing literature.

The title and short title say the SPF proteins are conserved while in the abstract it is written that the authors “think the SPF proteins may be conserved” (line 34). I would avoid using the term ‘conserved’ in the titles unless nucleotide/protein sequences of SPFs can be compared, and a clear threshold is provided between ‘non-conserved’ vs. ‘conserved’.

The novelty of indicating the presence of SPF in non-mental gland tissue in plethodontids is already interesting enough although this is not the first publication on this, but first for K. koreana. It is unnecessary to call it a pheromone or conserved to make it more interesting when in fact there is uncertainty.

ABSTRACT:

27-28: Should be => **“To examine localization patterns of proteins of the ancient sodefrin precursor-like factor (SPF) pheromone system in K. koreana, an antibody… “** – I find it correct to use ‘proteins of the SPF ancient pheromone system’ and not call it “the ancient SPF pheromone in K. koreana”. Protein pheromones in salamanders have been shown to relatively rapidly evolve, so although amphibians use proteins from the ancient SPF pheromone system, their pheromones are expected to be species- or even population- specific (at least in terms of sequences) and thus not ancient. Please note that species-specificity is in the original definition of a pheromone. Although there can be cross reaction to related species’ pheromones, one cannot expect Karsenia and Desmognathus to share a common, ancient and conserved pheromone without extraordinary evidence. I am sure the authors did not mean to bring such a message, but to a reader, especially non-expert, it might seem like that as it is currently stated.

31: Should include ‘tail base glands’ that are mentioned in the title: **“males: mental glands and two types of dorsal tail base glands – caudal courtship glands and dorsal granular glands. SPF…”** – as it is now, there is no connection between ‘tail base glands’ in the title and ‘caudal courtship glands’ and ‘dorsal granular glands’ in the abstract which is confusing for someone screening abstracts.

33-34: Maybe something like **“Due to immunoreaction specificity, we think the SPF proteins of K. koreana and D. ocoee are related and are likely used as pheromones in both species.”** Instead of “Based on these data, we think the SPF protein may be conserved between K. koreana and D. ocoee.” – I would avoid saying that SPF protein is conserved between Kk & Do. This can be used in a discussion, but it does not seem suitable for the abstract.

INTRODUCTION:

49: “rocky slopes” seems redundant here. Either describe the habitat in more detail, especially if you find it relevant for the research or remove this since it is too general. E.g. knowing whether it is a terrestrial or aquatic species would bring much more relevant information to a reader than “rocky slopes”.

49-52: Strange sentence, please reformulate, especially the use of ‘but’.

53: The goal seems to have been to test for the presence of SPF proteins in the gland tissues and not “study K. koreana pheromone composition”. Please correct and be aware that a pheromone can be 1 molecule or several molecules acting together (e.g. PRF & PMF in some plethodontids; SPF & persuasins in salamandrids). It would not be surprising if there are also other SPF proteins in K. koreana that did not react to the D. ocoee SPF antibody because they were more distantly related, e.g. they could still be produced even by the caudal courtship glands despite your method could not detect them. The goal is clearly stated in the abstract **“To examine localization patterns of SPF…” based on D. ocoee SPF antibody.**

56-68: Majority of plethodontids, including the D. ocoee, use **non-olfactory diffusion delivery (Houck et al., 2008)** in which the male scratches the female’s dorsum with his teeth and secrets the pheromone from mental gland onto female’s wound. It also happens that this type of delivery is **associated with the use of SPF** and is thus relevant for K. koreana. How the SPF reaches the receptors or where are the receptors is not known. The authors here describe olfactory delivery typical for members of a genus of North American plethodontids that predominantly use PRF and PMF proteins as pheromone components and not SPF. I find it very surprising that the authors would avoid focusing or even mentioning diffusion delivery while it seems **most relevant**. After all, D. ocoee’s SPF that served to produce SPF antibodies used in this research is transferred to a female using diffusion delivery and not the described olfactory delivery. This is described in detail in Houck et al. 2008 (with an image of courtship enclosed) that is not only cited in this paper multiple times, but also one of the co-authors of this paper is the last author of Houck et al., 2008. Please include diffusion delivery and discuss the relevance for K. koreana throughout the article.

65-68: Please reformulate. Plethodontids do not exist since the origin of amphibians and for the SPF, as far as it is known, they are suspected to be involved in sexual communication in frogs and confirmed in salamanders but not in caecilians or any other taxon outside frog-salamander clade. The two listed references do not and cannot corroborate the 360 million years origin of amphibians claim. Numbers and extraordinary statements are easily picked up and cited by others (e.g. unsuspected review authors), so please, as experts in the field, refrain making extraordinary statements without extraordinary evidence.

69: “Despite the variation…” – strange and completely unnecessary, please reformulate, especially since only olfactory delivery was explained earlier.

If the article is aimed at a wider audience, I would suggest explaining in the Introduction the following terms: **courtship**, **courtship pheromones**, **courtship glands** and **examples** in salamanders.

78-79: again, the goal changed, please see comment under line 53 and correct. There are no guarantees that the D. ocoee SPF protein antibody will bind all SPF and allow to check for SPF expression. The result can only be presence vs. absence of SPF proteins that bind to D. ocoee SPF antibody.

MATERIALS AND METHODS

84: I suggest “Tissue from three K. koreana and three D. ocoee male specimens were received…”

92-93: missing line between paragraphs (as used in 100, 132 & 150)

93-99: There should be an explanation on why was this kind of staining used, what is it expected to stain and what does it reveal? See comment 233-234.

103: Please replace ‘pheromones’ with ‘proteins’ if content is unknown or not pure. Currently, only the SPF has a proposed courtship pheromone function in D. ocoee (see comment under ‘Title’). Other components are either unknown, or do not have a proposed pheromone function. It is likely that there is one multicomponent courtship pheromone consisting of 1 or more SPFs and maybe other proteins or peptides. Since there is currently no evidence for multiple pheromones in D. ocoee mental glands, please refrain of using terms like ‘pheromones extracted from D. ocoee mental glands’.

104: Please cite ‘Houck, 2008’ correctly. See comment under 275. No brackets are necessary around the reference in this case – ‘… from Houck et al., 2008’.

134: The abbreviation IHC for the term ‘immunohistochemistry’ is redundant since it is used only a couple of times in the article (see comment under line 186).

RESULTS

160: should be ‘plethodontid species’

164: should be ‘posses’ unless it refers to males whose mental glands have been removed for the study, in which case it should be stated that “males used in this study possessed…”

165-166: strange sentence, please reformulate to make it more easily understandable, something like “the submandibular region was isolated and staining was used to identify the glands according to literature (**REF**) (Fig. 1B)”

166: please see comment under 164

164-172: The whole paragraph should be more nicely rewritten with more structure.

174-177: Please make clear to the readers in the figure legend what are the A, B and C. E.g. which organism for which letter. I get that Do is (C) so I can only suppose that (A) & (B) are Kk – this must be made clear. The images should give more context. Again, it is intended for wider readership. So, the skin can be indicated and then different parts of the mental gland tissue – e.g. secreting cells and lumen of the secretory ducts. Colours should be explained in the legend – where does a reader need to focus, is it to the red, pink, yellow, purple, white... As it is now, all the Mg (mental gland) arrows point to the empty white lumens of the glands which looks strange. It may be helpful to define a mental gland throughout the article. Is every follicle an individual mental gland or does all mental gland tissue in a male form one mental gland? A mental gland is quite a big, pronounced gland cluster in breeding males, so reducing it to a single follicle is confusing. Something like calling a nephron a kidney to illustrate my remark. I understand that this distinction is often not clear with amphibian skin glands (unlike with kidney) as they are often dispersed, rather than compact, but it may be worth it to keep it more clear for the readers. E.g. calling individual follicles mental gland tissue and the entire cluster of follicles mental glands. A control image (no staining) of the gland tissue would be appreciated, although not necessary if well explained in the legend.

181: A bracket after B is missing. After the missing bracket I suggest starting a new sentence beginning with “Contrary…”, “Conversely…”, “Inversely…” or something similar to gain clarity.

185: should be “SPF proteins” instead of “SPF pheromones” (see 27-28 comment)

186: There are only a few mentions of immunohistochemistry in the article so I find the use of abbreviation “IHC” ineffective.

187-188: The statement does not appear scientific in its current form. I suggest: “This recombinant SPF protein was synthesised using the nucleotide sequence of the major D. ocoee SPF protein isoform…”

188-189: If the antibody is made based on D. ocoee SPF protein from mental gland, how can staining D. ocoee mental gland with this antibody ensure antibody specificity? Please clarify.

194: same as 186

200: “mental glands” under (A) should be removed. Now mental glands are in both (A) and (B) which is not the case. The authors could make it clear in the Fig. 3 legend that it is about negative control. In that way the reader immediately knows what it is about. Most people will probably only check the images, so please make the figure legends as clear and informative as possible.

DISCUSSION

207: Since it is not sure what was stained (the staining does not seem to be specific as it stains both Kk and Do tissue), please use a more general term such as “presence of SPF proteins” instead of “presence of SPF isoform”.

206-210: The sentence is too long, confusing, with too much information and thus difficult to read. It seems like the same SPF protein was identified in Kk and Do, which cannot be claimed without further evidence, e.g. on the sequence level. Please make this whole paragraph clear and accurate for the readers.

212-222: Here the authors should use the opportunity to draw comparisons between Kk and Do. Do they expect them to have similar courtship behaviour? If yes, why? Different known, plethodontid pheromone systems (PRF, PMF, SPF…) and different courtship modes can be discussed from an evolutionary perspective, but relevant to Kk and Do. With a special emphasis to diffusion delivery that is currently completely omitted from the study, while being most relevant. This is a relatively straightforward study, so packing it in a well written context worthy of journal article is crucial and expected from experts in the field.

217-218: If there is any reference to Kk skin glands and sexual behaviour, this is the place to cite those references (there were none so far). “Not much is known…” without references does not offer any information to the reader and may be even wrong. What if there is nothing known at all? If possible, please draw parallels between Kk and Do skin glands and behaviour.

221-222: There could also be an option that SPF is not present.

224-230: Currently best paragraph of the discussion. That the antibody was made based on denatured SPF protein is also interesting to mention here.

233-234: The statement: **“This is the first report of PAS-negative glands being positive for a pheromone.”** needs context. Is it the first time in plethodontids, salamanders, amphibians, chordates, animals…? What does PAS stain and why is it relevant for pheromone glands? For which kind of pheromones is it relevant - all pheromones or only of a specific type? Why is PAS used and what does it stain should also be explained in the ‘Materials and methods’ section.

238-239: Glands “are empty during scent-marking” should be something like “glands empty during scent marking” or “glands get empty after scent-marking”. If glands ‘are empty’ during scent-marking, then no scent mark should come out of them.

240-241: This last sentence might be acceptable for a thesis or a popular article, but it does not seem suitable for a scientific journal. This should be written in a scientific manner.

243-249: Please rewrite the conclusion in a more structured and scientific manner. It is good that the authors mention the possibility of presence of other SPF proteins not bound by their Do SPF antibody.

REFERENCES:

There are references in reference list not cited in the text, e.g. Promislow, 1987; Staub & Paladin, 1997; Larson et al. 2006... Please check and correct.

275: Houck, 2008 seems like a key reference of this article and must include all the 6 co-authors, not just Houck. In the text this reference is referred to either as (Houck et al. 2008) – 67, 213 or (Houck, 2008) - 104. Please list all the co-authors in the reference list and cite accordingly throughout the text.

6. PLOS authors have the option to publish the peer review history of their article (what does this mean?). If published, this will include your full peer review and any attached files.

Reviewer #1: No

Reviewer #2: No

Reviewer #3: No

---

## [Author Response · Author response to Decision Letter 0]

7 Jul 2023

I have attached a PDF of this information that has my comments in red so that it is easier to read.

Reviewer's Responses to Questions

Comments to the Author

1. Is the manuscript technically sound, a d do the data support the conclusions?

Reviewer #1: Yes

Reviewer #2: Yes

Reviewer #3: Yes

2. Has the statistical analysis been performed appropriately and rigorously?

Reviewer #1: N/A

Reviewer #2: N/A

Reviewer #3: N/A

3. Have the authors made all data underlying the findings in their manuscript fully available?

Reviewer #1: Yes

Reviewer #2: Yes

Reviewer #3: Yes

4. Is the manuscript presented in an intelligible fashion and written in standard English?

Reviewer #1: Yes

Reviewer #2: Yes

Reviewer #3: Yes

5. Review Comments to the Author

REVIEWER #1:

This study investigates the production of potential protein pheromones in glandular tissue of the only Asian plethodontid salamander. The authors show via immunohistology that two of three glands produce SPF – an ancient pheromone known from many amphibians.

The manuscript is well written, the experiments are nicely conducted and the results are nicely presented and comprehensively discussed. The methods used by the authors are a very good solution, especially when only preserved tissue is available.

My only larger issue is the structuring of the introduction (which should be relatively easy to fix). Besides that, I only have a few smaller issues/comments.

Main issue:

Introduction:

The introduction starts very abrupt, right away introducing the study species and only then talking about salamander pheromones. I think a bit of restructuring would give the introduction a better flow and also make it more interesting for a broader readership. I suggest starting with a few broader sentences regarding pheromones and chemical communication in general, then go over to amphibian chemical communication, then introduce SPF as the most ancient pheromone found in amphibians, and then go into detail about plethodontid courtship and pheromones, before introducing the species of interest and your hypothesis with regard to this species.

“...I suggest starting with a few broader sentences regarding pheromones and chemical communication in general…”

Great point. The introduction does start abruptly and does not flow as well as it could. As a result, the article may not capture the interest of PLOS ONE’s wide viewership. To fix this, we modified the manuscript in these ways (while also capturing other comments):

To draw in the journal's broad readership, we started with chemical signaling and why it is interesting to scientists. 

We then discussed what differentiates a pheromone from any substance/chemical and the distinction between pheromone and pheromone-candidates.

After that, we discussed a familiar example of pheromones to many readers - insects.

See lines 54-61 for these changes.

“...then go over to amphibian chemical communication, then introduce SPF as the most ancient pheromone found in amphibians, and then go into detail about plethodontid courtship and pheromones…”

Here is how we changed our introduction accordingly:

As you mentioned in later comments, SPF is only a hypothesized pheromone in frogs, so we tied this in with the last paragraph by mentioning “pheromone-candidates”. This is helpful because it also sets up discussions that occur later in the article on why we are hesitant to classify SPF definitively as a pheromone in mental glands or caudal courtship glands of K. koreana, as no K. koreana courtship has been observed, let alone studied. Thank you for the suggestion!

We then mention SPF and its presence in frogs. 

We describe why SPF was proposed as a potential courtship pheromone in frogs in the first place by mentioning SPF as a confirmed pheromone in salamanders and newts. 

Discuss plethodontid salamander courtship behavior in relation to SPF.

See lines 63-84 for these changes.

“…before introducing the species of interest and your hypothesis with regard to this species…”

Thank you for the feedback. Another reviewer pointed out that we could provide more information regarding K. koreana’s discovery and why it is interesting to members of the field. We included this to help readers better understand the importance of this study (and improve the flow). 

Hopefully, our changes not only improve the flow but help contextualize the study for the wider audience.

See lines 86-106.

Further comments:

- line 66-68: the way this sentence is written it sounds a bit like SPF is an ancient pheromone that today only exists in plethodontids. I know this is not what you mean, but maybe you can add that it has been shown to have a pheromone function in many salamander families (for review see CSiV-chapter from Bossuyt et al. 2019) and has been even found in many frogs (where, however, proof that it is a pheromone is still lacking)

Yes, good point - a non-expert would have likely interpreted this meaning. In our edited version, we introduce SPF as a candidate pheromone in frogs and describe it as a pheromone in salamanders and newts. 

See lines 65-70.

- line 103: since Richard C. Feldhoff is a co-author in this paper, I think it is not necessary to mention this here (you could instead maybe add a statement at the end of the manuscript stating the different author contributions)

Removed his name here, thank you!

- line 96: …the PAS reaction (stains neutral carbohydrates),…

Changed. Thank you! Also, to improve the flow and ease of understanding for the reader, we included why we conducted these stains: to identify the different gland types in K. kroeana. We also provided more context for why the PAS stain is important (mental glands and caudal courtship glands stain similarly). 

See lines 90-93 and 117-121. 

- line 99: what does dimedone in this context?

I added a sentence after dimedone’s mention to describe its purpose. I also included more procedural detail about this reagent (1 hour at 60 degrees C after periodic acid treatment). 

See lines 122-123. 

- line 104/106-7: …following the methods from Houck (2008) …. adapted from Wilburn et al. (2018) and antibody purification methods from Wilburn and Feldhoff (2019).

Changed. Thank you! This was updated with the citation system, so these references are numbers now. 

- line 158-162: this part is not necessary here

Thank you for the suggestion! Removed.

- only when seeing Fig. 2 and reading line 232 in the discussion I realized that the caudal courtship glands and the granular glands are directly adjacent to each other. It would be helpful to already describe their positions better in the introduction or (even better) show a picture/drawing of a salamander where the different glands are marked/pointed out

This is an excellent idea. To give the reader more context, we included information about what glands are in the introduction to prime them for the images in the results section. We also buffed out the figure captions and figures to be more robust, walking the reader through the different components of a gland. 

See lines 98-100 and the figure captions.

- 243ff: this might exceed the framing of the discussion, but (maybe for future studies) it could be worth considering that some of the plethodontid glands release volatile pheromones, as shown for example for reed frogs and mantellids?

I was unsure how to frame this in the discussion, so I choose not to include this. The broad readership may find it out of place as we did not discuss volatile pheromones at all in the paper. However, this is a very interesting point! 

All in all, this is a very nice study and manuscript and I congratulate the authors on their interesting findings!

Thank you, and we appreciate your feedback!

REVIEWER #2

DeBruin et al., Presence of the conserved sodefrin precursor-like factor pheromone in mental and

dorsal tail base glands in the plethodontid salamander, Karsenia koreana

I enjoyed reading this article, and I think it makes a nice contribution to the understanding of the evolutionary history of plethodontid pheromones. I have mostly minor comments aimed at improving clarity below. On a more major note, you might consider adding some phylogenetic/evolutionary history context to the Discussion. In the Introduction, you argue that the pheromone SPF is conserved within the family, and describe some of the phylogenetic/evolutionary history of this (Lines 64-68), but you do not circle back to the conservation of SPF in the Discussion now that you have discovered it in K. koreana. It would be helpful to put SPFs discovery in K. koreana in context of the entire family. I think it would also help your argument in Lines 243-249 to discuss how the isolation in both evolutionary time and geographic distance between K. koreana and other plethodontids might explain why a D. ocoee SPF anitbody might not pick up a long-diverged SPF variant in the Ccg.

Thank you for the feedback! Due to some restructuring in the introduction and comments from other reviewers, I took out some of the phylogenetic discussion in the introduction. Reviewer #3 pointed out that while we see evidence of SPF immunoreactivity, we need sequence-level data to claim that SPF is “conserved,” so it is difficult to make any phylogenetic/evolutionary claims without it. However, I agree that putting K. koreana in context with its family is an important part of this paper. To address this, we included the following:

We discuss how K. koreana’s potential courtship behaviors by relating this species to other plethodontid salamanders. See lines 224-237.

We point out that using an antibody from D. ocoee SPF hints at the structural and functional similarity of this protein between D. ocoee and K. koreana. See lines 239-245.

We also bring in some phylogenetic evidence that K .koreana and D. ocoee’s most recent common ancestor, about 50 million years ago, may account for the antibody not recognizing SPF proteins in the caudal courtship glands of K. koreana (as you suggested). See lines 258-265.

Minor Comments:

Line 49: I might be misremembering, but isn't K. koreana the only extant plethodontid throughout all of Asia? If so, you can make this statement stronger than "the only plethodontid in Korea."

Thank you for pointing this out, we updated the manuscript to accurately represent this. See lines 102-104.

Line 69-70: "another highly conserved behavior" - replace "another" with "a", and add "courtship" before behavior to be clear. You just talked about behaviors with variation, and a conserved pheromone, so tail straddle walk is the first highly conserved behavior you're discussing.

Due to some restructuring requested by other reviewers, this part was removed entirely. However, the tail straddling walk is still mentioned as well as its role in plethodontid courtship mainly in the discussion. See lines 247-256.

Line 84: add "three male" before Desmognathus, OR add "each" after specimens, to be clear that you had three of each species.

Changed. Thank you for the suggestion! See line 109.

Lines 158-162: I don't know that the "as mentioned previously" is necessary. You could shorten the first two sentences of the Results into one sentence (example below), and combine this paragraph with the next paragraph for a more concise Results section.

Ex. "The skin glands of plethodontid salamanders have been thoroughly described in several species, however characterization in K. koreana has been limited. Here, we examined three skin gland types in K. koreana:..."

We eliminated these sentences from the results at the request of another reviewer. Thank you!

Line 181: In the Figure 2 Legend, you are missing close parenthesis after (B

Fixed. Due to some restructuring of the results section, we have changed the order of the figures. 

Lines 185-198: Were you expecting the Ccg/Gg to show presence of SPF? That is, do D. ocoee or other plethodontids show SPF in the Ccg/Gg or just in the Mg? Your D. ocoee reference (Fig. 1C.) is just of the Mg. If you did stain Ccg and Gg in D. ocoee, you might add that to Figure 2 to mirror how you laid out Figure 1.

“...Were you expecting the Ccg/Gg to show presence of SPF? That is, do D. ocoee or other plethodontids show SPF in the Ccg/Gg or just in the Mg?”

We were expecting to see SPF in Ccg but not in Gg. The Ccgs are not confirmed as pheromone-producing glands, as those studies have been not been conducted yet. However, there is evidence that these glands play a role in the tail straddling walk. We clarified this in the introduction (86-95) and the beginning of the discussion (215-236).

“Your D. ocoee reference (Fig. 1C.) is just of the Mg. If you did stain Ccg and Gg in D. ocoee, you might add that to Figure 2 to mirror how you laid out Figure 1.”

We did not stain D. ocoee Ccgs and Mgs unfortunately, but we would like to do a follow-up study looking at SPF in Ccgs of other plethodontid salamanders. 

I was initially confused on what Figure 3 was showing until I reread lines 196-198 and the legend a few times. It might be worth adding something to the legend similar to the sentence on Lines 196-198 to make it clear that this is a control for the SPF antibody staining in Figs. 1 and 2. Indeed, you have a very clear statement on the importance of this image at the end of the first Discussion paragraph (Lines 209-210).

Thank you, we clarified that this is a negative control in the figure title and kept the sentences you mentioned in the discussion.

Line 241: Change "–could they be related to SPF?" to "and they could be [structurally?] related to SPF". Where I have structurally, put whatever qualifier(s) make sense, since "related" by itself is extremely broad.

Great point. Due to some changes, we made it more specific by stating, “there are likely highly conserved regions within these proteins.” (239-241).

Line 247-249: For this next step, you would need live individuals from which to extract the secretions, correct? I would state this outright for the benefit of those who don't work on glandular chemistry (like myself).

Great point. Changed! This is one of the final sentences of the discussion. 

REVIEWER #3: 

General comment:

Given the straightforward study goal and most of the co-authors that are well known in the field I expected a better written article with little investment in the review process. However, this is not the case. The English is generally at a good level but with strange expressions and statements to the point that I felt like correcting a bachelor or master thesis and not a scientific article submitted to a respectful journal by experts in the field. There is a major oversight of not mentioning the diffusion delivery in plethodontids to provide more context to the study, but this can be easily corrected. Since most of the authors are well-known experts in the research field, I do not see any reason why this study could not have a solid context that it and the journal deserve. As it is written now, I more get the impression that the article is written to ‘get it out there as fast as possible’ than to provide high quality material for the readers interested in the subject.

In detail:

TITLE:

The title suggests that SPF is a pheromone in Karsenia salamanders while no such studies have been conducted. Furthermore, although there is indication that SPFs from mental glands act as a pheromone in North American plethodontids (based on 20-25 kDa band of fraction D with unknown content of multiple bands tested on Desmognathus ocoee by Houck et al. in 2008), the SPF mode of action is completely unknown, i.e. where are female receptors, how does SPF reach them and how does it affect female behaviour? The production of SPF in mental glands and potential use of SPF as a pheromone in the tail-base glands in Asian plethodontids is the biggest novelty of this research, but it also calls for caution calling SPFs from tail-base glands pheromones before empirical studies have been conducted. I would suggest calling SPF proteins “pheromone-candidates” instead, something in the sense of:

“Sodefrin precursor-like factor pheromone candidates present in both mental and dorsal tail base glands in the plethodontid salamander, Karsenia koreana”

Thank you for pointing out this oversight, the title has been updated to reflect this suggestion. We have also ensured that oversights like this are not made in the manuscript. Any mention of SPF in the context of K. koreana is referred to as proteins involved in the SPF pheromone system, SPF proteins (not pheromones), pheromone-candidates, or similar, as you suggested. 

The title indicates ‘tail base glands’ while the abstract mentions ‘dorsal granular glands’ and ‘caudal courtship glands’ which I find quite confusing. There should be consistency in the naming of the tissues throughout the article according to existing literature.

Clarified that dorsal tail base glands refer to caudal courtship glands and granular glands in the introduction and abstract (see line 36 and lines 89-91).

Regarding consistency with the existing literature, the terms “caudal courtship glands”, and “granular glands” and their associated abbreviations, “Ccg” and “Gg”, have been used previously to describe this type of gland (Example 1, Example 2, Example 3). 

Furthermore, both caudal courtship glands and mental glands have been used to describe these same types of glands in Karsenia koreana specifically. 

The title and short title say the SPF proteins are conserved while in the abstract it is written that the authors “think the SPF proteins may be conserved” (line 34). I would avoid using the term ‘conserved’ in the titles unless nucleotide/protein sequences of SPFs can be compared, and a clear threshold is provided between ‘non-conserved’ vs. ‘conserved’.

Changed. Thank you for pointing this out!

The novelty of indicating the presence of SPF in non-mental gland tissue in plethodontids is already interesting enough although this is not the first publication on this, but first for K. koreana. It is unnecessary to call it a pheromone or conserved to make it more interesting when in fact there is uncertainty.

This is an excellent point, the title has been changed to reflect this suggestion. 

ABSTRACT:

27-28: Should be => “To examine localization patterns of proteins of the ancient sodefrin precursor-like factor (SPF) pheromone system in K. koreana, an antibody… “ – I find it correct to use ‘proteins of the SPF ancient pheromone system’ and not call it “the ancient SPF pheromone in K. koreana”. Protein pheromones in salamanders have been shown to relatively rapidly evolve, so although amphibians use proteins from the ancient SPF pheromone system, their pheromones are expected to be species- or even population- specific (at least in terms of sequences) and thus not ancient. Please note that species-specificity is in the original definition of a pheromone. Although there can be cross reaction to related species’ pheromones, one cannot expect Karsenia and Desmognathus to share a common, ancient and conserved pheromone without extraordinary evidence. I am sure the authors did not mean to bring such a message, but to a reader, especially non-expert, it might seem like that as it is currently stated.

Thank you for the suggestion! We have changed this in the manuscript (lines 32-33).

31: Should include ‘tail base glands’ that are mentioned in the title: “males: mental glands and two types of dorsal tail base glands – caudal courtship glands and dorsal granular glands. SPF…” – as it is now, there is no connection between ‘tail base glands’ in the title and ‘caudal courtship glands’ and ‘dorsal granular glands’ in the abstract which is confusing for someone screening abstracts.

Changed! Thank you for the suggestion (line 36).

33-34: Maybe something like “Due to immunoreaction specificity, we think the SPF proteins of K. koreana and D. ocoee are related and are likely used as pheromones in both species.” Instead of “Based on these data, we think the SPF protein may be conserved between K. koreana and D. ocoee.” – I would avoid saying that SPF protein is conserved between Kk & Do. This can be used in a discussion, but it does not seem suitable for the abstract.

Thank you for this suggestion. We have updated the manuscript accordingly (lines 38-40). We also bring back this hypothesis into the discussion (lines 224-237).

INTRODUCTION:

Another reviewer pointed out that the introduction was clunky and required some restructuring. Some of the suggestions provided here were moved/removed in the revision. 

49: “rocky slopes” seems redundant here. Either describe the habitat in more detail, especially if you find it relevant for the research or remove this since it is too general. E.g. knowing whether it is a terrestrial or aquatic species would bring much more relevant information to a reader than “rocky slopes”.

Removed as it was not relevant to the research. We substituted it with a greater emphasis on the irregularity of finding this plethodontid in Asia to help the paper flow and engage the wide readership (lines 103-105).

49-52: Strange sentence, please reformulate, especially the use of ‘but’.

This sentence was removed in the process of reorganizing the introduction. 

53: The goal seems to have been to test for the presence of SPF proteins in the gland tissues and not “study K. koreana pheromone composition”. Please correct and be aware that a pheromone can be 1 molecule or several molecules acting together (e.g. PRF & PMF in some plethodontids; SPF & persuasins in salamandrids). It would not be surprising if there are also other SPF proteins in K. koreana that did not react to the D. ocoee SPF antibody because they were more distantly related, e.g. they could still be produced even by the caudal courtship glands despite your method could not detect them. The goal is clearly stated in the abstract “To examine localization patterns of SPF…” based on D. ocoee SPF antibody.

Thank you for your suggestion, we updated the language in the introduction to accurately depict that pheromones can be one or a combination of both. 

We discuss your suggestion in the discussion - the possibility that because D. ocoee and K. koreana are distantly related, the D. ocoee antibody does not recognize SPF in K. koreana Ccgs. 

56-68: Majority of plethodontids, including the D. ocoee, use non-olfactory diffusion delivery (Houck et al., 2008) in which the male scratches the female’s dorsum with his teeth and secrets the pheromone from mental gland onto female’s wound. It also happens that this type of delivery is associated with the use of SPF and is thus relevant for K. koreana. How the SPF reaches the receptors or where are the receptors is not known. The authors here describe olfactory delivery typical for members of a genus of North American plethodontids that predominantly use PRF and PMF proteins as pheromone components and not SPF. I find it very surprising that the authors would avoid focusing or even mentioning diffusion delivery while it seems most relevant. After all, D. ocoee’s SPF that served to produce SPF antibodies used in this research is transferred to a female using diffusion delivery and not the described olfactory delivery. This is described in detail in Houck et al. 2008 (with an image of courtship enclosed) that is not only cited in this paper multiple times, but also one of the co-authors of this paper is the last author of Houck et al., 2008. Please include diffusion delivery and discuss the relevance for K. koreana throughout the article.

This is a crucial oversight, we have replaced our discussion of olfactory delivery with a discussion of the diffusion delivery system. Thank you for picking up on this error. Dedicated sentences in the discussion and introduction have been added to discuss diffusion delivery (lines 69-84 and 224-237). 

65-68: Please reformulate. Plethodontids do not exist since the origin of amphibians and for the SPF, as far as it is known, they are suspected to be involved in sexual communication in frogs and confirmed in salamanders but not in caecilians or any other taxon outside frog-salamander clade. The two listed references do not and cannot corroborate the 360 million years origin of amphibians claim. Numbers and extraordinary statements are easily picked up and cited by others (e.g. unsuspected review authors), so please, as experts in the field, refrain making extraordinary statements without extraordinary evidence.

Thank you for catching this and bringing it to light. We have adjusted these sentences to reflect that SPF is confirmed in salamanders but hypothesized in frogs. We also removed the 360 million years claim.

69: “Despite the variation…” – strange and completely unnecessary, please reformulate, especially since only olfactory delivery was explained earlier.

Removed! Thank you.

If the article is aimed at a wider audience, I would suggest explaining in the Introduction the following terms: courtship, courtship pheromones, courtship glands and examples in salamanders.

Courtship: “In this study, courtship is defined as the behavior to maintain reproductive actions between mating partners and does not refer to initial mate attraction” (lines 70-71).

Courtship pheromones: We defined the term “receptivity” to address this, as courtship pheromones impact the receptivity of courtship behavior as defined by Houck et al. 2008 (lines 80-81).

Courtship glands: First, We introduced glands in general and what they look like to prime the reader for the results (lines 76-77). We also then go on to explain the importance of the mental gland and caudal courtship glands. Also, we include a brief description of granular glands (where they are located in relation to caudal courtship glands) to prime the reader.

We discuss D. ocoee transdermal pheromone delivery as an example of pheromone communication in salamanders for the journal's wide readership.

78-79: again, the goal changed, please see comment under line 53 and correct. There are no guarantees that the D. ocoee SPF protein antibody will bind all SPF and allow to check for SPF expression. The result can only be presence vs. absence of SPF proteins that bind to D. ocoee SPF antibody.

Changed!

MATERIALS AND METHODS

84: I suggest “Tissue from three K. koreana and three D. ocoee male specimens were received…”

Changed!

92-93: missing line between paragraphs (as used in 100, 132 & 150)

Changed the heading format to match journal requirements which fixed the issue. 

93-99: There should be an explanation on why was this kind of staining used, what is it expected to stain and what does it reveal? See comment 233-234.

Discussed the stain’s relevance in the introduction, and how mental glands and caudal courtship glands stain similarly. We also described how this stain was used in the materials and methods (see lines 89-93).

103: Please replace ‘pheromones’ with ‘proteins’ if content is unknown or not pure. Currently, only the SPF has a proposed courtship pheromone function in D. ocoee (see comment under ‘Title’). Other components are either unknown, or do not have a proposed pheromone function. It is likely that there is one multicomponent courtship pheromone consisting of 1 or more SPFs and maybe other proteins or peptides. Since there is currently no evidence for multiple pheromones in D. ocoee mental glands, please refrain of using terms like ‘pheromones extracted from D. ocoee mental glands’.

Changed!

104: Please cite ‘Houck, 2008’ correctly. See comment under 275. No brackets are necessary around the reference in this case – ‘… from Houck et al., 2008’.

Changed with the citation format of the journal.

134: The abbreviation IHC for the term ‘immunohistochemistry’ is redundant since it is used only a couple of times in the article (see comment under line 186).

Changed!

RESULTS

160: should be ‘plethodontid species’

Changed (and moved to the discussion). 

164: should be ‘posses’ unless it refers to males whose mental glands have been removed for the study, in which case it should be stated that “males used in this study possessed…”

Changed! Thank you. 

165-166: strange sentence, please reformulate to make it more easily understandable, something like “the submandibular region was isolated and staining was used to identify the glands according to literature (REF) (Fig. 1B)”

Changed! We also included more information about this in the introduction and materials and methods (89-93, 117-126)

166: please see comment under 164

Changed.

164-172: The whole paragraph should be more nicely rewritten with more structure.

Took out the first part of the paragraph and moved to the discussion. Reformulated. 

174-177: Please make clear to the readers in the figure legend what are the A, B and C. E.g. which organism for which letter. I get that Do is (C) so I can only suppose that (A) & (B) are Kk – this must be made clear. The images should give more context. Again, it is intended for wider readership. So, the skin can be indicated and then different parts of the mental gland tissue – e.g. secreting cells and lumen of the secretory ducts. Colours should be explained in the legend – where does a reader need to focus, is it to the red, pink, yellow, purple, white... As it is now, all the Mg (mental gland) arrows point to the empty white lumens of the glands which looks strange. It may be helpful to define a mental gland throughout the article. Is every follicle an individual mental gland or does all mental gland tissue in a male form one mental gland? A mental gland is quite a big, pronounced gland cluster in breeding males, so reducing it to a single follicle is confusing. Something like calling a nephron a kidney to illustrate my remark. I understand that this distinction is often not clear with amphibian skin glands (unlike with kidney) as they are often dispersed, rather than compact, but it may be worth it to keep it more clear for the readers. E.g. calling individual follicles mental gland tissue and the entire cluster of follicles mental glands. A control image (no staining) of the gland tissue would be appreciated, although not necessary if well explained in the legend.

“Please make clear to the readers in the figure legend what are the A, B and C. E.g. which organism for which letter. I get that Do is (C) so I can only suppose that (A) & (B) are Kk – this must be made clear.”

Thank you for the suggestion - we changed this in the manuscript.

“The images should give more context. Again, it is intended for wider readership. So, the skin can be indicated and then different parts of the mental gland tissue – e.g. secreting cells and lumen of the secretory ducts. Colours should be explained in the legend – where does a reader need to focus, is it to the red, pink, yellow, purple, white... As it is now, all the Mg (mental gland) arrows point to the empty white lumens of the glands which looks strange.”

Thank you for the feedback. We included labels on the mental glands and a more robust first caption to help the reader identify the important parts of the glands. We also included a description of gland morphology similar to the introduction in this figure caption. We connected these parts to different colors as well to highlight where the reader should look in all 3 figures. 

“It may be helpful to define a mental gland throughout the article.”

Defined in the introduction and in the figure caption as well. We discussed glands in the context of this study and how they can be identified. 

181: A bracket after B is missing. After the missing bracket I suggest starting a new sentence beginning with “Contrary…”, “Conversely…”, “Inversely…” or something similar to gain clarity.

Changed!

185: should be “SPF proteins” instead of “SPF pheromones” (see 27-28 comment)

Removed this part entirely with your earlier comment.

186: There are only a few mentions of immunohistochemistry in the article so I find the use of abbreviation “IHC” ineffective.

Removed!

187-188: The statement does not appear scientific in its current form. I suggest: “This recombinant SPF protein was synthesised using the nucleotide sequence of the major D. ocoee SPF protein isoform…”

Removed form the results and used it in the methods.

188-189: If the antibody is made based on D. ocoee SPF protein from mental gland, how can staining D. ocoee mental gland with this antibody ensure antibody specificity? Please clarify.

We found that this sentence in general was confusing, so we decided to remove it, as it did not help us further our argument.

194: same as 186

Done!

200: “mental glands” under (A) should be removed. Now mental glands are in both (A) and (B) which is not the case. The authors could make it clear in the Fig. 3 legend that it is about negative control. In that way the reader immediately knows what it is about. Most people will probably only check the images, so please make the figure legends as clear and informative as possible.

Done!

DISCUSSION

207: Since it is not sure what was stained (the staining does not seem to be specific as it stains both Kk and Do tissue), please use a more general term such as “presence of SPF proteins” instead of “presence of SPF isoform”.

Changed, see lines 224-225.

206-210: The sentence is too long, confusing, with too much information and thus difficult to read. It seems like the same SPF protein was identified in Kk and Do, which cannot be claimed without further evidence, e.g. on the sequence level. Please make this whole paragraph clear and accurate for the readers.

We changed our language here to be more accurate and clearer to readers (239-240). Instead of “same protein” we said “perhaps highly conserved regions.”

212-222: Here the authors should use the opportunity to draw comparisons between Kk and Do. Do they expect them to have similar courtship behaviour? If yes, why? Different known, plethodontid pheromone systems (PRF, PMF, SPF…) and different courtship modes can be discussed from an evolutionary perspective, but relevant to Kk and Do. With a special emphasis to diffusion delivery that is currently completely omitted from the study, while being most relevant. This is a relatively straightforward study, so packing it in a well written context worthy of journal article is crucial and expected from experts in the field.

We added the relevant plethodontid diffusion delivery system in D. ocoee compared to K. koreana. Also, we juxtapose this information with phylogenetic data that shows K. koreana shares a most recent common ancestor with Hydromantes plethodontids. Hydromantes males, while observed to rub their mental gland on female dorsal skin, also rub their mental gland on the female’s nares. We use both pieces of evidence to highlight K. koreana’s potential similarities and differences to members of the plethodontid family.

217-218: If there is any reference to Kk skin glands and sexual behaviour, this is the place to cite those references (there were none so far). “Not much is known…” without references does not offer any information to the reader and may be even wrong. What if there is nothing known at all? If possible, please draw parallels between Kk and Do skin glands and behaviour.

There are no references to K. koreana skin glands and sexual behavior. There have only been studies on gland morphology, but no studies on courtship. We changed the manuscript to reflect this.

221-222: There could also be an option that SPF is not present.

Added!

224-230: Currently best paragraph of the discussion. That the antibody was made based on denatured SPF protein is also interesting to mention here.

Added that the antibody was made from denatured SPF to highlight this interesting finding.

233-234: The statement: “This is the first report of PAS-negative glands being positive for a pheromone.” needs context. Is it the first time in plethodontids, salamanders, amphibians, chordates, animals…? What does PAS stain and why is it relevant for pheromone glands? For which kind of pheromones is it relevant - all pheromones or only of a specific type? Why is PAS used and what does it stain should also be explained in the ‘Materials and methods’ section.

Defined PAS in the materials and methods as well as the introduction sections so that what it stains and why it matters is clear. 

Removed this sentence as it did not add anything to the discussion.

238-239: Glands “are empty during scent-marking” should be something like “glands empty during scent marking” or “glands get empty after scent-marking”. If glands ‘are empty’ during scent-marking, then no scent mark should come out of them.

We ended up removing this part from the discussion.

240-241: This last sentence might be acceptable for a thesis or a popular article, but it does not seem suitable for a scientific journal. This should be written in a scientific manner.

Changed! We now have a brief conclusion paragraph that is more scientific.

243-249: Please rewrite the conclusion in a more structured and scientific manner. It is good that the authors mention the possibility of presence of other SPF proteins not bound by their Do SPF antibody.

Created a cleaner conclusion that summarizes the study in full.

REFERENCES:

There are references in reference list not cited in the text, e.g. Promislow, 1987; Staub & Paladin, 1997; Larson et al. 2006... Please check and correct.

275: Houck, 2008 seems like a key reference of this article and must include all the 6 co-authors, not just Houck. In the text this reference is referred to either as (Houck et al. 2008) – 67, 213 or (Houck, 2008) - 104. Please list all the co-authors in the reference list and cite accordingly throughout the text.

Citations have been changed and are now updated according to the journal requirements. Thank you for pointing this out! 

6. PLOS authors have the option to publish the peer review history of their article (what does this mean?). If published, this will include your full peer review and any attached files.

Do you want your identity to be public for this peer review? For information about this choice, including consent withdrawal, please see our Privacy Policy.

Reviewer #1: No

Reviewer #2: No

Reviewer #3: No

---

## [Editor Report · Decision Letter 1]

17 Jul 2023

Presence of sodefrin precursor-like factor pheromone candidates in mental and dorsal tail base glands in the plethodontid salamander, *Karsenia koreana*

PONE-D-22-29926R1

Dear Dr. DeBruin,

We’re pleased to inform you that your manuscript has been judged scientifically suitable for publication and will be formally accepted for publication once it meets all outstanding technical requirements.

Kind regards,

M. Caitlin Fisher-Reid, Ph.D.

Guest Editor

PLOS ONE

Additional Editor Comments (optional):

I was originally Reviewer 2 on the first draft of this manuscript, and PLOS ONE asked me to be the guest editor. I have now thoroughly read the other reviews, the revised manuscript, and the authors' response to reviewers, and I think the authors have done a wonderful job restructuring the manuscript for a the more general PLOS ONE audience as well as satisfactorily addressing all other reviewer comments. Congratulations!
---

## [Editor Report · Acceptance letter]

20 Jul 2023

PONE-D-22-29926R1 

Presence of sodefrin precursor-like factor pheromone candidates in mental and dorsal tail base glands in the plethodontid salamander, *Karsenia koreana*

Dear Dr. DeBruin:

I'm pleased to inform you that your manuscript has been deemed suitable for publication in PLOS ONE. Congratulations! Your manuscript is now with our production department. 

Kind regards, 

on behalf of

Dr. M. Caitlin Fisher-Reid 

Guest Editor

PLOS ONE